# The Gender Gap Tracker: Using Natural Language Processing to measure gender bias in media

**Fatemeh Torabi Asr**[1], **Mohammad Mazraeh**[1], **Alexandre Lopes**[2], **Vagrant Gautam**[1], **Junette Gonzales**[1], **Prashanth Rao**[1], **Maite Taboada**[1]*

1 Discourse Processing Lab, Department of Linguistics, Simon Fraser University, Burnaby, Canada, 2 Research Computing Group, Simon Fraser University, Burnaby, Canada

* mtaboada@sfu.ca

**Data Availability Statement:** The data was downloaded from public and subscription websites of newspapers, under the 'fair dealing' provision in Canada's Copyright Act. This means that the data

## Abstract

We examine gender bias in media by tallying the number of men and women quoted in news text, using the Gender Gap Tracker, a software system we developed specifically for this purpose. The Gender Gap Tracker downloads and analyzes the online daily publication of seven English-language Canadian news outlets and enhances the data with multiple layers of linguistic information. We describe the Natural Language Processing technology behind this system, the curation of off-the-shelf tools and resources that we used to build it, and the parts that we developed. We evaluate the system in each language processing task and report errors using real-world examples. Finally, by applying the Tracker to the data, we provide valuable insights about the proportion of people mentioned and quoted, by gender, news organization, and author gender. Data collected between October 1, 2018 and September 30, 2020 shows that, in general, men are quoted about three times as frequently as women. While this proportion varies across news outlets and time intervals, the general pattern is consistent. We believe that, in a world with about 50% women, this should not be the case. Although journalists naturally need to quote newsmakers who are men, they also have a certain amount of control over who they approach as sources. The Gender Gap Tracker relies on the same principles as fitness or goal-setting trackers: By quantifying and measuring regular progress, we hope to motivate news organizations to provide a more diverse set of voices in their reporting.

## Introduction: The Gender Gap in media and in society

Women's voices are disproportionately underrepresented in media stories. The Global News Monitoring Project has been tracking the percentage of women represented in mainstream media since 1995, when it was 17%. Twenty years later, in 2015, it had increased to only 24%, with a worrisome stalling in the previous decade [1]. At this rate, it would take more than 70 years to see 50% women in the media, a true reflection of their representation in society.

can be made available only for private study and/or research purposes, and not for commercial purposes. As such, the data will be made available upon request and after signing a license agreement. Contact for data access: Maite Taboada (mtaboada@sfu.ca) or Carla Graebner, Data Librarian at Simon Fraser University (data-services@sfu.ca). The code is available on GitHub under a GNU General Public License (v3.0). The authors of this paper are the creators of the code and own the copyright to it: https://github.com/sfu-discourse-lab/GenderGapTracker A light-weight version of the NLP module is also made available for processing one 804 article at a time: https://gendergaptracker.research.sfu.ca/apps/textanalyzer.

**Funding:** We are grateful for funding and in-kind contributions from the following units at Simon Fraser University: the Big Data Initiative; the Office of the Vice-President, Research & International; and the Office of the Dean of Arts and Social Sciences. Funding was provided by grants to M. Taboada from the Social Sciences and Humanities Research Council (Insight Grant 435-2014-0171) and the Natural Sciences and Engineering Research Council of Canada (Discovery Grant 261104).

**Competing interests:** The authors have declared that no competing interests exist.

The underrepresentation of women is pervasive in most areas of society, from elected representatives [2–5] and executives [3, 6, 7] to presidents and faculty in universities [5, 8, 9]. Women are also underrepresented in political discussion groups [10]. It is, therefore, not entirely surprising that news stories mostly discuss and quote men: Many news stories discuss politicians and business executives, drawing on expert opinion from university professors to do so. Perversely, in many stories where women are overrepresented, it is because they are portrayed as having little or no agency, as in the case of victims of violence [11–14] or politician's spouses [15]. During international gatherings like G7/G8 or G20 summits, a set of stories often discuss the parallel meetings of spouses with a focus on their attire, and humorously commenting on cases when the lone man joins activities clearly planned for wives only [16, 17]. Countless studies have pointed out *how* the representation of women in media is different; e.g., [18–24]. In our project, we first tackle the question of *how much* of a difference there is in the representation of women; cf. [25].

Not a great deal of progress seems to have been made since Susan Miller found, in 1975, that photos of men outnumbered photos of women by three to one in the pages of the *Washington Post*, and by two to one in the *Los Angeles Times*. Among the more than 3,600 photos that Miller studied, women outnumbered men only in the lifestyle section of the two papers [26].

Most previous studies of gender representation in media have performed manual analyses to investigate the gap. Informed Opinions, our partner organization in this project, carried out a study in 2016, analyzing 1,467 stories and broadcast segments in Canadian media between October and December 2015, to find that women were quoted only 29% of the time [27]. The work was laborious and intensive. Similarly, the enormous effort of the Global News Monitoring Project is only possible thanks to countless volunteers in 110 countries and many professional associations and unions around the globe. Thus, it only takes place every five years. Shor et al.'s [24] study of a historical sample of names in 13 US newspapers from 1983 to 2008 found that the ratio went only from 5:1 in 1983 to 3:1 by the end of the period. It seems to be stubbornly stuck at that level. A recent analysis of news coverage of the COVID-19 pandemic [28] used a mix of manual and automatic methods and found that men were quoted between three and five times more often than women in the news media of six different countries.

The causes and solutions to the underrepresentation of women in society in general and in news articles in particular are too complex to discuss in this paper (but see [29–33]). We focus here on the first step in any attempt at change: an accurate characterization of the current situation. Just like a step tracker can motivate users to increase their physical activity, we believe that the Gender Gap Tracker can motivate news organizations to bring about change in areas they have control over. It is obvious that, if a news story requires a quote from the Prime Minister or a company's president, the journalist does not have a choice about the gender of those quoted. Journalists, however, do have control over other types of sources, such as experts, witnesses, or individuals with contrasting viewpoints.

Indeed, when journalists keep track of their sources and strive to be more inclusive, both anecdotal and large-scale evidence show that parity is, in fact, possible. Ed Yong, staff writer for *The Atlantic* who covers science news, reported that keeping track of his sources was the simple solution to ensure gender parity in his articles [34]. Ben Bartenstein, who covers financial news for Bloomberg, improved the gender ratio in his sources by keeping lists of qualified women and tracking the sources in his stories [35]. The BBC's 50:50 project (https://www.bbc.co.uk/5050) also uses strategic data collection and measurement to achieve 50% women contributing to BBC programs and content.

It is with this goal in mind—of motivating news organizations to improve the ratio of people they quote—that the Gender Gap Tracker was born. The Gender Gap Tracker is a

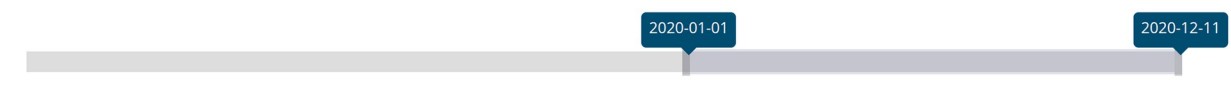

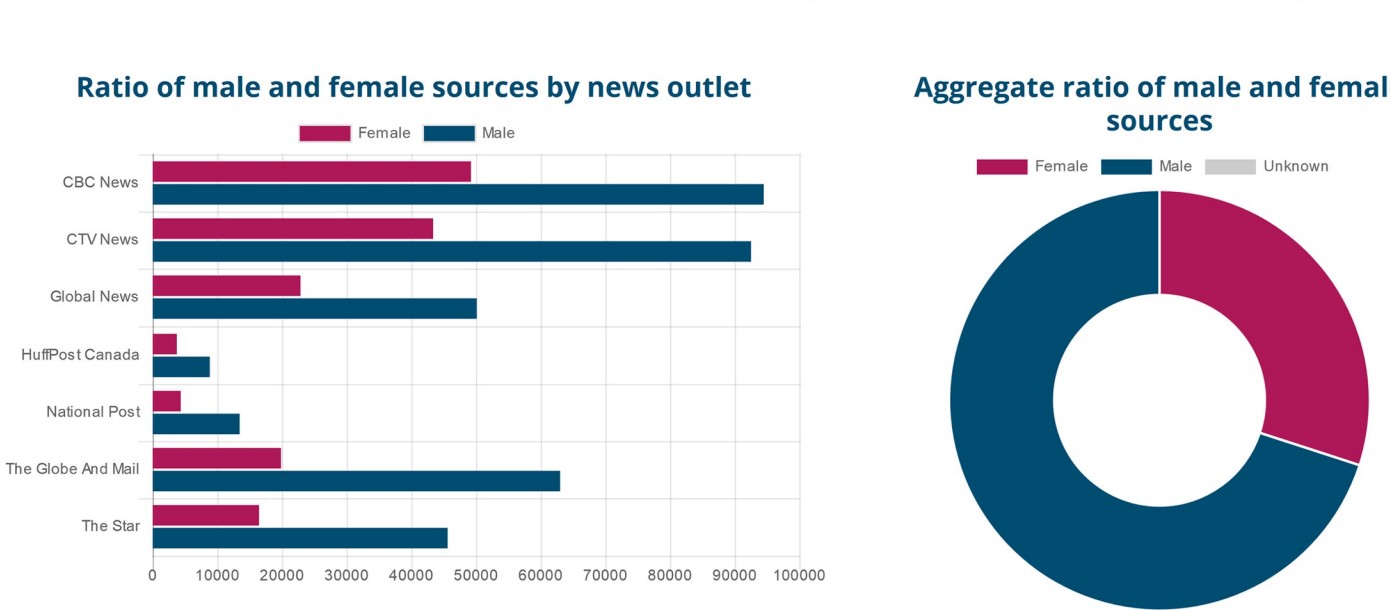

**Fig 1. The Gender Gap Tracker online dashboard page.** Reprinted from https://gendergaptracker.informedopinions.org/ under a CC BY license, with permission from Informed Opinions, original copyright 2018.

collaboration between Informed Opinions (https://informedopinions.org), a non-profit organization dedicated to amplifying women's voices in media, and Simon Fraser University, through the Discourse Processing Lab (http://www.sfu.ca/discourse-lab) and the Big Data Initiative (https://www.sfu.ca/big-data).

We harness the power of large-scale text processing and big data storage to collect news stories daily, perform Natural Language Processing (NLP) to identify who is mentioned and who is quoted by gender, and show the results on a public dashboard that is updated every 24 hours (https://gendergaptracker.informedopinions.org). The Tracker monitors mainstream Canadian media, seven English-language news sites (a French Tracker is in development), motivating them to improve the current disparity. By openly displaying ratios and raw numbers for each outlet, we can monitor the progress of each news organization towards gender parity in their sources. Fig 1 shows a screenshot of the live page. In addition to the bar charts for each organization and the doughnut chart for aggregate values, the web page also displays a line graph, charting change over time (see Fig 2 below).

For the two years since data collection started on October 1, 2018 until September 30, 2020, the average across the seven news outlets is 29% women quoted, versus 71% men, with a negligible number of unknown or other sources. We have, however, observed an increase in the number of women quoted between the first and the last month in that period, from 27% in October 2018 to 31% in September 2020. Some of that increase can be directly attributed to an increase in the quotes by public health officers during the COVID-19 crisis. It just so happens that a large number of those public health officers across Canada are women [36]. We report some of the analyses and insights we are gathering from the data in the section Analysis and observations.

In this paper, we describe the data collection and analysis process, provide evaluation results and a summary of our analysis and observations from the data. We also outline other

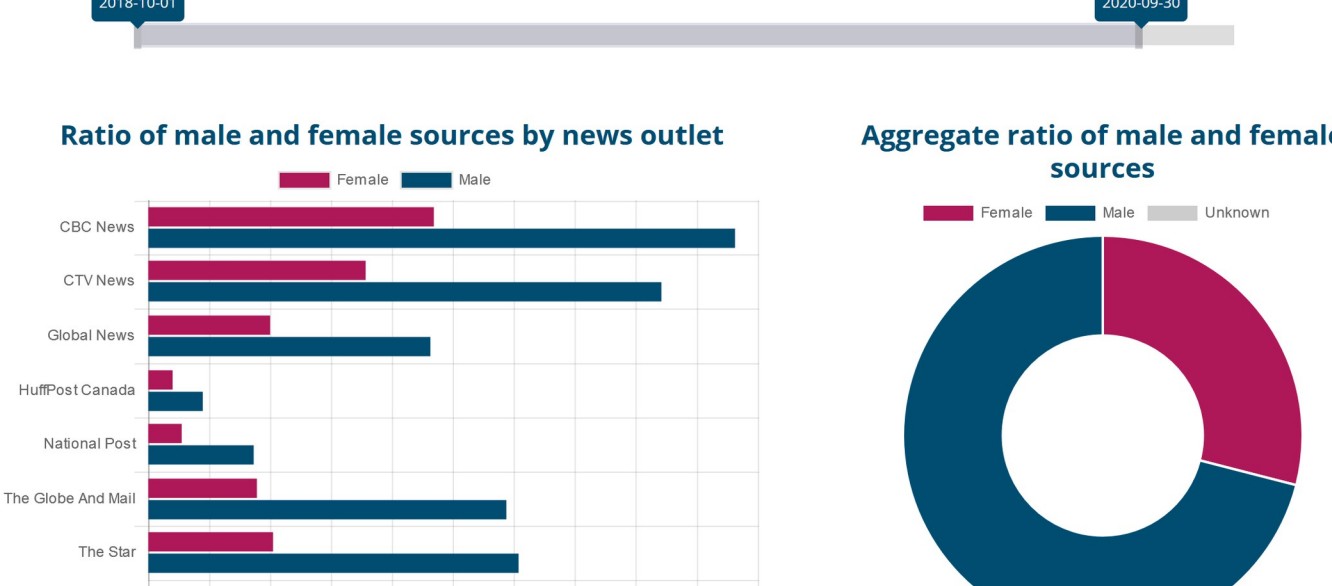

The line graph makes it easier to compare how often each news outlet quoted women over time. Its default mode is the most recent 3-month period, but when a specific date range is selected on the slider, the line graph will automatically update to reflect the chosen period.

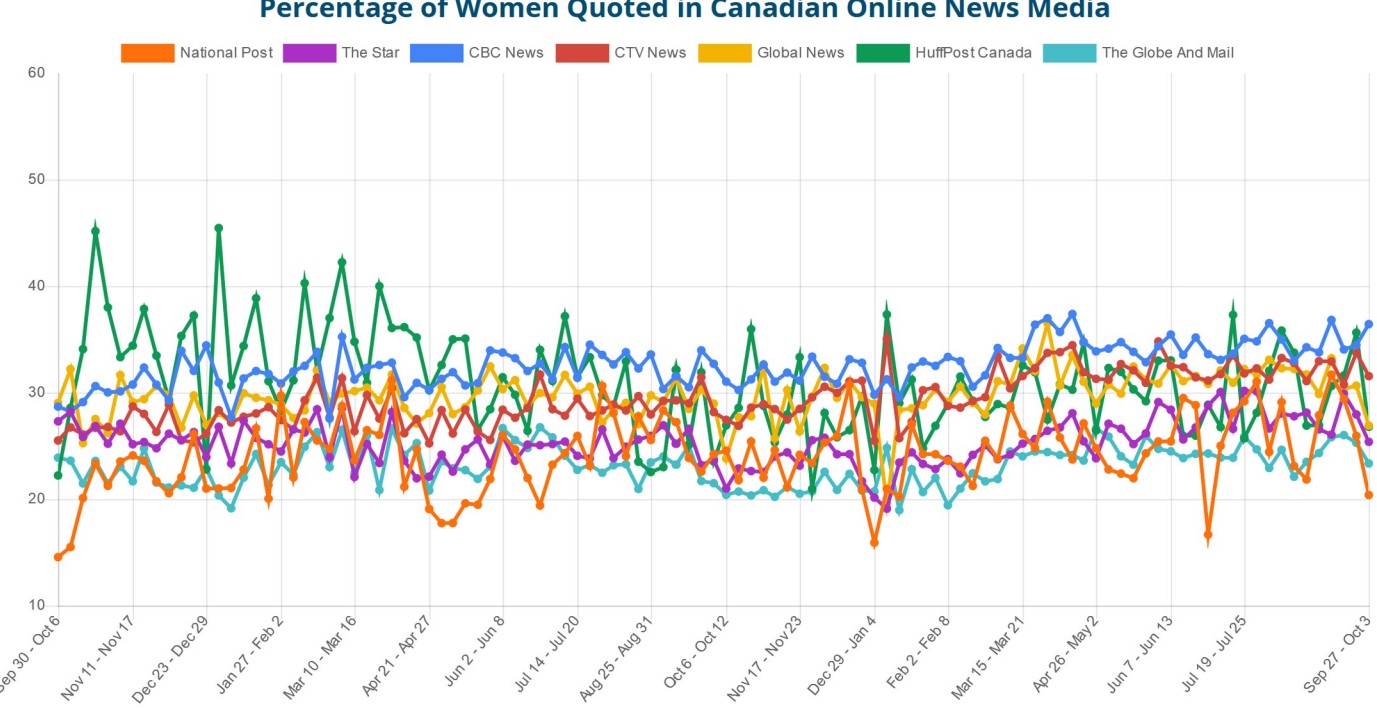

**Fig 2. Counts and percentages of male vs. female sources of opinion across seven news outlets.** Dates: October 1, 2018 to September 30, 2020. Female sources constitute less than 30% of the sources overall. *CBC News* (blue line) and *HuffPost Canada* (green line) show a better gender balance compared to other outlets; *The Globe and Mail* (light blue) and *The National Post* (orange) are at the bottom, quoting women less than 25% of the time. Reprinted from https://gendergaptracker. informedopinions.org/ under a CC BY license, with permission from Informed Opinions, original copyright 2018.

potential uses of the tool, from quantifying gender representation by news topic to uncovering emerging news topics and their protagonists. We start, in Related work, with a review of existing literature on quotation patterns, extracting information from parsed text, and potential biases in assigning gender to named entities. We then provide, in Data acquisition and NLP processing pipeline, a high-level description of the data acquisition process and how we deploy NLP to extract quotes, identify people, and predict their gender. More detail for each of those steps is provided in the S1 Appendix. Throughout the development of the Gender Gap Tracker, we were mindful of the need for accuracy, in both precision and recall of quotes, but also in terms of any potential bias towards one of the genders (e.g., disproportionately attributing names or quotes to one gender). In order to ensure that the Gender Gap Tracker provides as accurate a picture as possible, we have performed continuous evaluations. We describe that process in the section on Evaluation. The section Analysis and observations answers the most important questions that we posed at the beginning of the project: Who is quoted, in what pro-portions? We add more nuanced analyses about the relationship between author gender and the gender breakdown of the people those authors quote. Finally, Conclusion offers some reflections on the use of the Gender Gap Tracker as a tool for change, also discussing future improvements and feature additions.

Before delving into the technical aspects of the Gender Gap Tracker and the insights it provides about the gender gap in media, we would like to acknowledge that the language we choose to describe people *matters* and that the terms we use are simplifications of a complex reality. We use 'women' and 'men' and 'female sources' and 'male sources', implying a binary opposition that we know is far from simple. Gender is more nuanced than that. We know, at the same time, that lack of gender representation in many aspects of society is a reality. Our goal is to quantify that lack of representation by using language and the traditional associations of names and pronouns with men and women. We discuss this issue in more detail in the section on on Gender prediction and gender bias in Natural Language Processing.

## Related work

The Gender Gap Tracker involves the application of different insights and research findings in linguistics and Natural Language Processing. To our knowledge, there is no comparable project extracting both direct and indirect quotes on a continuous basis. Because so many different research fields are involved, it is challenging to provide a succinct summary of related existing work. We have focused our survey in this section on three aspects that have informed our work the most: descriptions of direct and indirect speech in linguistics, prediction of gender based on names in text, and extraction of dependency structures and quotes in Natural Language Processing to make a connection between entities and quotes.

### Reported speech

Reported speech is a recreation, or a reconstruction, of what somebody said in a certain context [37]. Note that even though we refer to it as reported *speech*, the concept applies equally to quotes from written text, such as a press release [38]. The structure is also used to recreate thought (*I thought "Okay. What am I gonna do?"*) or even action (*I was like "[choking/gagging sound]"*), especially in colloquial language [39, p. 44]. Vološinov [40] characterized reported speech as both 'speech within speech' and 'speech about speech'.

It is this nature of reported speech as the recreation of an event, whether involving speech or not, that makes it so important in interaction and in narrative. Goddard and Wierzbicka [41] propose that, regardless of typological differences in how it is expressed across the world's

languages, reported speech is fundamental to human society: Our environment is largely made up of other people's utterances in our stories, dreams, memories, and thoughts. Talk is certainly fundamental to cognition; more specifically, however, it is often talk about talk that "binds groups and communities together" [41, p. 173]. See also Vološinov [40] and Goddard and Wierzbicka [42].

In linguistics, a distinction is drawn between two types of reported speech: direct and indirect speech. Direct speech involves direct quotation of somebody's exact words, typically enclosed in quotation marks in writing. With indirect speech, we report on those words, perhaps altering the exact original formulation, and involving deictic shifts, i.e., changes in tense and point of view [38]. Thus, the direct speech *"I've chosen to start wearing a mask," Mr. Trudeau said* becomes *Mr. Trudeau said that he had chosen to start wearing a mask* in indirect form, with a change from *I* to *he* and *'ve chosen* to *had chosen*. The distinction may be labelled as direct vs. indirect quotation, or direct vs. indirect report [43]. In English and many other languages, it is generally understood that direct speech is used when the intention is to reproduce the speaker's words verbatim, that is, to be faithful not only to the content of the message, but also to the form in which it was uttered [44–46]. We will be using the term 'reported speech' for any recreation of what somebody said, as a broad term including both direct and indirect speech.

Reported speech has played an important role in our common cultural stock, including oral narrative and written fiction. We have progressively used it more and more as a form of evidence. Consider the so-called Miranda warning in the United States: Upon arrest, a suspect is told that anything they say may be used as evidence against them. Citations in scientific articles are also a form of reported speech as evidence. We cite or paraphrase other scientists' words as part of a scientific argument, and as part of the dialogue we engage in as researchers [47]. Reported speech, especially in its direct version, features in news discourse as a direct reproduction of somebody's exact words, as a safeguard against interpretation by the reporter in the form of indirect speech. (Note also that we refer to journalists as *reporters*, signalling their role in telling us the news.) The use of reported speech as evidence in news articles is what makes it such an interesting object of study. By identifying who is quoted in news articles, we capture whose words are considered important and worthy of repetition.

As we will see in Quotation extraction below, our analysis focuses on patterns of quotation typically found in news articles: quotes with a matrix clause, whether as direct or indirect speech, and direct quotes that appear in their own sentence (floating quotes). A lively debate in linguistics tries to elucidate whether reported speech is a syntactic, a semantic, a pragmatic, or a paralinguistic phenomenon [38, 39, 42, 48–51]. While reported speech probably requires a syntactic, semantic, and pragmatic analysis for a full account, here we use a structural approach and rely on NLP tools rooted in syntactic patterns to identify and extract quoted material, the reporting verb (the verb introducing the reported speech), and the speaker (or source) of the quote.

## Extracting quotes with Natural Language Processing

Reported speech, both direct and indirect, features specific syntactic structures that can be identified through automatic parsing. In direct speech, the presence of quotation marks, together with the presence of a reporting verb, signals a quote. For indirect speech, it is the reporting verb plus a specific syntactic structure, the dependent clause, that points to the presence of reported speech. The most reliable way to find that information, and to find the beginning and end of quotes, is to first create a parse tree or a dependency tree of the structure of the text.

The slightly different flavours of automatic parsing in NLP all result in a reading of the structure of sentences in constituents, with dependency structures identified either implicitly (through tree structure) or explicitly [52]. The focus of our attention in those structures are the complement clauses of reporting verbs.

Much of the research on dependency structures, especially for reported speech, involves the Penn Treebank, the Penn Discourse Treebank [53], and related collections of annotated news articles that are widely used in computational linguistics research. Early in the development of the Penn Discourse Treebank, it was clear that the discourse relations involved in reported speech needed to be addressed, as they featured prominently in the news articles present in the corpus. Consequently, a great deal of attention was paid to annotating attribution and its features in the original corpus, including source, level of factuality, and scope [54]. An extension of the annotations, the PARC corpus of attribution relations [55], contains a more fine-grained annotation with more relations, which can be used to train machine learning systems to detect quotations [56, 57]. We do not follow a machine learning approach here, as we believe not enough annotations are available for the wide range of reported speech types that we have encountered.

An approach that also relies on parse trees of sentences is that of van Atteveldt et al. [58], who extract the text of quotes from news reports. They compare this to a baseline and find that, although there are errors in the parse tree, a syntactic parsing method outperforms a baseline which relies on word order. Likewise, a method using a mix of parse trees and regular expressions is deployed by Krestel et al. [59] to identify both direct and indirect speech in news text.

In a large-scale approach similar to ours, but using rules, Pouliquen et al. [60] identify direct quotes (those surrounded by quotation marks) in news reports from 11 languages. This research led to the pioneering Europe Media Monitor (http://emm.newsexplorer.eu/), which tracks news events, top stories, and emerging themes in the news of over 70 countries (but with a focus on Europe). The quotation extraction, however, seems to have been discontinued in recent versions of the tool.

Our approach is medium-scale, in that it concentrates on Canadian English-language news sources, but is comprehensive enough in the sphere of the Canadian media landscape that trends in gender representation can be gleaned. By using reliable parsing information, we are confident that we detect the majority of quotes in different formats, covering both direct and indirect speech. To our knowledge, this is the most extensive quote analysis performed on a continuous basis.

## Gender prediction and gender bias in Natural Language Processing

The statistics that we are interested in the most, i.e., the gender breakdown of people quoted in the news, rely on accurate prediction of gender based on people's names. Although gender prediction based on this approach is straightforward and accuracy can be quite high, it is, like many other aspects of NLP, a site for potential bias.

Automatic gender prediction typically relies on the predictable gender associations of people's first and sometimes last names. For English-speaking countries, a common source of these associations is the US census and the Social Security Administration, where names are mapped to their most frequent sex association at birth (https://www.ssa.gov/oact/babynames/). Clearly, this is a problematic practice, as it assumes that gender is binary, that sex and gender have perfect correlation, and that people's names are accurate predictors of their sex or gender. We acknowledge and respect the complex nature of this matter, and we are open to further refinements of our approach, as discussions are underway at many levels.

For instance, the US census is considering how to best capture sexual orientation and gender identity [61].

The other main method for automatic gender prediction is the entity-based approach where a label is given based on the individual, i.e., an association of a first-last name combination with a specific person and their public gender identity. This is feasible with public figures, as their gender can be extracted from online resources such as Wikidata or HathiTrust [62]. As we will section on Identifying people and predicting their gender, we apply both first name and first-last name methods by extracting information from online services.

We acknowledge that gender is non-binary, and that there are different social, cultural, and linguistic conceptualizations of gender. For this project, we rely on self- and other-identification of gender through names in order to classify people mentioned and quoted as female, male, or other. In English, the third person singular pronoun *he* encodes male gender and *she* encodes female gender. First names tend to be used distinctively by persons of different genders. We recognize that some people prefer a gender neutral pronoun (*they*) and that some people adopt or have been given names that are not strongly associated with one gender (e.g., *Alex*). We are aware that, because our technical approach is based on a simplified view of gender prediction, it glosses over the many possible gender identities, does not quantify the bias of our tools towards traditional white Western names (which tend to be overrepresented in training data), or intersectionality. This is just a start in the conversation about representation in the media, and we tackle this first attempt through the encoding of gender in language, which in English is mostly binary. All our statistics and analyses include a categorization of gender in three parts: female, male, and other. The latter includes cases where the gender of a person, based on their name, is unknown (because the name is used for both genders), or non-binary (because the person identifies as non-binary).

One issue that we would like to point out here is the inherent bias in many standard NLP tools, datasets, and methods. While we have not fully measured how such biases affect our results, we do bear them in mind when making generalizations about the data. For instance, Garimella et al. [63] show that different syntactic patterns displayed by men and women can lead to different levels of accuracy in part-of-speech tagging and parsing. Therefore, if the parsing method we rely on has been trained on data primarily written by men and quoting men, it is quite possible then that its accuracy is lower when parsing and extracting quotes from women. Caliskan et al. [64] make a compelling case that implicit human biases are learned when using standard machine learning methods to extract associations from data. Among the biases Caliskan and colleagues found are associations of gender from names and careers (e.g., female names more associated with family than career words; more associated with the arts than with mathematics). Gender biases have also been found in coreference resolution [65, 66], visual semantic role labelling [67], and machine translation [68, 69].

In general terms, the type of bias that we are concerned about is what Blodgett et al. [70] term *representational harm*, specifically two types of representational harm: i) a difference in system performance for different social groups (different parsing accuracy for male and female voices); and ii) system performance that leads to misrepresentation of the distribution of different groups in the population (incorrect gender prediction that misrepresents the true proportion of men and women quoted). Ultimately, we are aware that these biases exist in text because they reflect inherent biases in society, and that attempts at minimizing them are not always successful [71]. We report error rates for our gender prediction process in the Evaluation section, and also make some observations in the Most frequent sources by category section about error rates for categories of people quoted. In general, we can say that our error rate is very low and that it does not seem to show bias.

## Data acquisition and NLP processing pipeline

This section provides a summary of the steps in acquiring data and processing it so that we can extract quotes, the people who said them, and the gender of those speakers (or sources). This is an overview of the process, which is described in much more detail in the S1 Appendix.

### Data scraping

Scraping public data from the web appears to be a simple task. We have found, however, that daily data scraping from heterogeneous sources is actually quite complex and requires customization of existing libraries. We had to deal with a variety of challenges arising from the different technologies, standards, and layouts used by the news outlet websites. This made it difficult to find a common pattern and write a script that could collect the data from different news outlets efficiently and in real time. The S1 Appendix contains further information on the techni-cal aspects of this process.

The final pipeline is a 24/7 online service composed of a set of scrapers in the background of the Gender Gap Tracker website. Each scraper is an individual process for a specific news outlet, scheduled to run twice a day, collect the daily publication of the target website, and store them in our central database. Each process takes between 5 and 30 minutes to execute each day, depending on the target outlet and the number of daily articles published, which tends to range from 800 to 1,500.

Once we have the article and all its metadata in the database, we move onto the Natural Language Processing piece of the pipeline, which involves extracting quotes, identifying people, and predicting their gender.

### Quotation extraction

To measure the gender representation gap in news media, we identify the number of men and women who are quoted in news articles; in other words, people who have not only been mentioned but have also seen their voices reflected in news. We consider both direct speech (surrounded by quotation marks) and indirect speech (*She stated that*. . .) to be quotations. We refer to the speaker of such quotes as a *source* in the news article. In order to identify sources, we first need to extract quotes from the news article text, to then align quoted speakers with the unified named entities that are gender-labelled through the procedure described in the next section. While reported speech in general may be described as a semantic—rather than a syntactic—phenomenon [48], from an NLP point of view, the most reliable mechanism to identify it is the syntactic structure of sentences. Based on study of the literature on reported speech and our initial study of the data, we separate quotes into two different types and apply different procedures to each: syntactic quotes and floating quotes.

What we refer to as syntactic quotes follow a structure whereby a framing or matrix clause, containing the identity of the speaker (the subject) and a reporting verb, introduces a reporting clause, containing the material being quoted [38]. They may be direct or indirect quotes, but they share a common syntactic structure. Such quotes can be identified by finding patterns in a syntactic or dependency parse of the text, as in Example (1), where the structure *Janni Aragon. . . says* introduces the content of what the speaker said.

(1).   Janni Aragon, a political science instructor at the University of Victoria, says research shows different adjectives are used to describe female leaders compared to male counterparts.

When multiple quotes by the same speaker are present in a news article, it is often the case that only one syntactic quotative structure is used, with subsequent quotes receiving their own

sentence or sentences, all in quotes, as in Example (2). The first quote contains a quotative verb and speaker (*Kim told*). The second quote, *The fact that*. . . is a separate sentence without a quotative verb. We label these cases 'floating quotes'.

(2).   "Honestly, it feels like we're living our worst nightmare right now" Kim told CTV News Friday. **"The fact that we are being accused right now of an unethical adoption is crazy."**

In the above example, the second sentence is a continuation of Kim's quotation. However, Kim's name is not mentioned as the speaker of the quote anymore. Readers understand implicitly that this second quotation is from the same person mentioned in the previous sentence. These are also referred to as *open quotations* [50]. Spronck and Nikitina [38] characterize them as 'defenestrated', because the framing or matrix clause that typically introduces reported speech is absent. We identify floating quotes by following the structure of the text and matching their speaker to the most recently mentioned speaker.

Using the above two procedures, we capture a variety of syntactic and floating quotes with their verb and speaker. We also introduced a heuristic system for detecting quotes that were initially missed by the syntactic process. Further details on how we extract each type of quote are provided in the S1 Appendix. The next step connects each of these quotations to an entity identified as a source, labelled by gender.

## Identifying people and predicting their gender

We apply gender prediction techniques not only to sources (i.e., the people quoted), but also to all the people mentioned in the text as well as the authors of the articles. Since the main goal of the study is tracking the gender gap, it is very important that the identification of people and gender predictions are performed as accurately as possible.

As a first step towards extracting mentions of people in text, we use Named Entity Recognition (NER), a commonly used procedure in NLP. Current NER techniques work fairly well on English data. These methods are statistical in nature, relying on large amounts of annotated data and supervised or semi-supervised machine learning models, with neural network models being the most commonly used models nowadays [72, 73].

We first extract only entity types tagged with the label PERSON. This excludes organizations and locations that may look like names of people (e.g., *Kinder Morgan* or *Don Mills*). We then proceed to entity clustering. The same person may be referred to in the same article with slightly different names or pronouns (e.g., *Justin Trudeau, the Prime Minister, Mr. Trudeau, he, his*). To unify these mentions into clusters, and ensure that we attribute quotes to the right person, we apply a coreference resolution algorithm, described in the S1 Appendix.

The coreference process results in a unique cluster for each person containing all mentions in the text that refer to that person. Thus, we can count the number of people mentioned in the text and move to the next step, i.e., predicting their genders.

For gender prediction of each unified named entity (cluster of mentions), we rely on gender prediction web services that use large databases to look up a name by its gender. Initially, we experimented with using pronouns to predict gender (*he* or *she*), but found that this method was not reliable, because not all clusters of reference to an individual include a pronoun (see S1 Appendix for details).

The gender prediction web services that we use perform lookups by first names only, based on databases of names and sex as assigned at birth, or lookups by first and last name, using information for that specific individual and how they are identified publicly. We also keep an internal cache of names that we have previously looked up. In addition, the cache contains

manual entries for names that we know are not available in public databases, or are incorrectly tagged by a gender service.

We apply the gender prediction algorithm to three different lists of names: people mentioned in the article, people quoted (who we refer to as *sources*), and authors of articles. This process, especially for authors of articles, involves extensive data cleaning (see S1 Appendix).

To match the name and gender of the speaker to quotes, we find the corresponding named entity for each extracted quote. In order to do so, we compare the character indices of a quote's speaker against the indices of each named entity mention in our unified clusters. If a mention span and a speaker span have two or more characters of overlap, we assume that the mention is the speaker and attribute the quote to the unified named entity (coreference cluster) of the mention. After trying to align all quotation speakers with potential named entities, there may still remain some quotes with speakers that could not be matched with any of the named entities. There are several categories of these cases, such as quotes with a pronoun speaker (e.g., *she said*) where the pronoun is still a singleton after all named entity and coreference cluster merging. Our current version of the software ignores these cases. We provide statistics on these and other missed cases in the evaluation section below.

## Evaluation

Evaluation of the system was continuous in the development phase, with each new addition and improvement being tested against the previous version of the system, and against manual annotations. Evaluation was carried out separately for each component (quote extraction, people and source extraction, and gender prediction) several times over the course of the project to test out new ideas and to enhance the system. In this section, we discuss the main annotation and results of our evaluation for the most recent release of the system, V5.3. Further details on a pilot annotation and the format of the manually-annotated dataset can be found in the S1 Appendix.

For evaluation, we selected 14 articles from each of the seven news outlets, for a total of 98 articles, chosen from months of recently scraped data at the time (December 2018-February 2019). We chose articles that were representative of the overall statistics according to our system, i.e., contained less than 30% female and more than 70% male sources (calculated based on the latest system release at the time of annotation). The articles were picked in a way that they were distributed across different days of the week and each was selected to have at least 3,000 characters.

We draw articles from our database, rather than using unseen data, for two reasons. First of all, since none of the processes involve supervised learning on this data, there is no risk that the system will have learned anything from the test data. The NLP methodology uses a combination of pre-trained language models (from spaCy), linguistic rules, and custom phrase matching. Thus, we can safely assume that any true positives captured during evaluation will generalize to the rest of our data as well. Second, we are primarily interested in how the system performs specifically on the data we are processing. While evaluation on news articles by other organizations may be useful, we are most of all interested in our performance on the data the Gender Gap Tracker collects daily.

An experienced annotator, who had participated in our pilot annotation, completed the data labelling. The annotations were then also validated and corrected when necessary by a second annotator.

For each of the 98 articles, we have a JSON file which contains an array of extracted quotes, verbs, and speakers, together with their character span indices in the text. We evaluate the output of our system by comparing it to these human annotations. To do so, first we need to align

**Table 1. Quote extraction evaluation on manually annotated data.**

| | Quotation content | | | Verb | Speaker |
|---|---|---|---|---|---|
| | Precision | Recall | F1-score | accuracy | accuracy |
| Easy match threshold (0.3) | 84.6% | 82.7% | 83.7% | 91.8% | 86.0% |
| Hard match threshold (0.8) | 77.0% | 75.2% | 76.1% | 93.1% | 86.9% |

the annotations with the extracted quotes. Let $q_a$ be the span of an annotated quote and $q_e$ the span of an extracted quote. The match between these two quotes is defined as:

$$score = \frac{len(q_a \cap q_e)}{len(q_a)} \tag{1}$$

For each annotated quote $q_a$, the best matching quote from among all extracted quotes is the one with the highest matching score, assuming the score is above a certain threshold. We experimented with 0.3 and 0.8 as easy and hard thresholds, respectively. We found that 0.3 captured a relatively large portion of each quote, and 0.8 captured the majority of the content. In the following example, the human annotated and automatically extracted quote spans are highlighted using italic and underlined text, respectively. The alignment score is 0.45, which is the ratio of the length of the overlapping portion (69 characters) to the overall length of the annotated span (153 characters).

(3). *"It's premature for us to make any sort of pronouncement about that right now, but <u>I can tell you this thing looks and smells like a death penalty case</u>"*.

After alignment, we examine how many of the quotes were correctly detected (true positives), how many were not detected (false negatives), and whether we have some non-quote sentences detected as quotes (false positives). With these numbers, we report the precision, recall, and F1-score of the system in Tables 1 and 2.

Table 1 shows the result of evaluating the quotation extraction code on the manually-annotated dataset. The first three columns of numbers reflect how well the system captures the quotation content span (according to each of the set threshold of overlap 0.3 and 0.8) and the last two columns show system accuracy on verb and speaker detection. We consider the verb to have been correctly detected if the verb extracted by the system has exactly the same span as the expert-annotated span for the verb of that quotation. In order to evaluate the speaker detection quality at the surface textual level, we apply a simple overlap threshold: If the system-annotated span for the speaker has at least one character overlap with the expert-annotated text span for the speaker, it will be accepted as a correct annotation. For example, if the system-annotated span was [12:17], corresponding to the string *Obama*, while the human-annotated span was [8:17], corresponding to the string *Mr. Obama*, the span overlap of five characters would mean they were considered the same speaker. Verb and speaker evaluations are applied only to the matched quotes (the quotations that are already passed as aligned

**Table 2. Entity extraction evaluation based on manually annotated data.**

| | Human annotation, *n* | System annotation, *n* | Precision | Recall | F1-score |
|---|---|---|---|---|---|
| Female people | 2,906 | 3,387 | 72.4% | 77.6% | 75.0% |
| Male people | 8,381 | 10,034 | 77.4% | 92.1% | 84.2% |
| Female sources | 1,442 | 1,104 | 94.6% | 64.6% | 76.8% |
| Male sources | 3,809 | 3,346 | 87.7% | 76.5% | 81.8% |

between system and expert based on the content span overlap). That is why the accuracy scores for Verb and Speaker in the table were higher when we used a stricter quote matching technique (hard match threshold).

## People and sources

The most important data point with respect to the goal of our project is the ratio of female and male sources. Therefore, we compare the raw number of people and sources of each gender extracted by our system against the corresponding numbers in the human expert annotation.

Furthermore, we would like to know how many of the people mentioned in the text were correctly detected and how many were missed. According to the annotation instructions, the most complete name of each person in the text needs to be provided by the annotators in the annotation files. We have the following arrays of names for each article: female people, male people, other/unknown-gender people, female sources, male sources, and other/unknown-gender sources. Using these manually annotated lists, we can calculate the number of entities our system detects and misses. We first convert all system- and expert-annotated entities in these lists to lowercase and trim the start/end space characters. Then we perform exact string matching on the elements of the arrays to calculate the precision, recall, and F1-score of each identification task. Note that this is a strict evaluation of the system performance and it is directly motivated by our goal to reveal the proportion of female and male sources in news publications.

Table 2 shows the results of entity matching between the system- and expert-annotated people and sources. We see better precision scores in detection of sources in comparison with people. The reason is that the quote extraction step narrows down the people list by filtering out the captured entities that were not quoted at all (so some errors such as location names tagged as people names would automatically be excluded). The recall measure shows the opposite trend: Recall is better for people than for sources. This is because the same narrowing down that improves the precision for sources results in an increase in the number of missed sources, thus negatively affecting recall.

One more interesting gender-related pattern we found was that, in general, we had better recall for male people mentioned and sources, compared to the female mentions and sources. This motivated us to take a closer look at the data and see whether there was any systematic bias in our entity recognition and/or gender recognition procedures, by carrying out a manual analysis.

## Manual analysis of top sources

In addition to the comparison to a full set of articles described above, we also checked the gender accuracy for the top sources in each of the 24 months between October 2018 and September 2020. The results are gratifyingly accurate: The overall error rate is 0.1%. Table 3 provides a breakdown of the error rate per gender (false positives). Note that we examined the top 100 male and female sources per month, but each of those people is quoted multiple times. As a consequence, the number of quotes examined is quite large (over 195,000). There are three aspects to highlight in Table 3:

- Considering that we are examining a constant number of sources per month (top 100 men and top 100 women quoted), it is clear that men are overrepresented in the dataset. That is, the top 100 men each month are quoted much more frequently than the top 100 women. We discuss this further in Analysis and observations.

**Table 3. Gender prediction accuracy for the top sources of opinion.**

| Number of quotes | | Error rate (false positives) | |
|---|---|---|---|
| Total quotes by men | 140,156 | Error rate for men | 0.1% |
| Quotes by men incorrectly identified | 147 | | |
| Total quotes by women | 55,149 | Error rate for women | 0.2% |
| Quotes by women incorrectly identified | 117 | | |
| | | Overall error rate | 0.1% |

- The error rate for quotes by women is higher. In our list of quotes by women, we see a higher proportion of names that were actually men (0.2%). That is, the system is more accurate in recognizing the gender of male names. This means that there probably is, in fact, a slightly lower number of quotes from women than our official statistics on the dashboard show, as more quotes are incorrectly attributed to women.

- Most of the errors in the female name list are names that are actually male names or ambiguous (*Ashley, Robin*). Most of the errors in the male name list are names that are actually not people's names. They include *Raymond James*, an investment firm, and *Thomas Cook*, a travel agency. We correct both types of errors on a regular basis, by adding information to our internal caches.

## Analysis and observations

In this section, we provide statistics on the data extracted from the seven news outlets, processed and tagged by the Gender Gap Tracker in the time frame of October 1, 2018 to September 30, 2020, 24 months of data and about 613,000 news articles. All numbers are based on the calculations of the Gender Gap Tracker version 5.3 (the most recently released version at the time of publication of this paper).

### Male vs. female sources

Fig 2 shows the statistics available on the Gender Gap Tracker dashboard online. The aggregated counts and ratios of female vs. male sources across different news outlets within the time interval of October 2018 to September 2020 are presented in the bar and the doughnut charts at the top. The bottom line graph shows the percentage of women quoted in the publications of each outlet week by week. Most numbers are in the range of 20 to 30 percent, meaning that women are consistently quoted far less often than men. While some outlets such as *Huffington Post* and *CBC News* are more gender-balanced than others, such as *The National Post* and *The Globe and Mail*, the numbers suggest that, overall, media outlets disproportionately feature male voices. This may be the result of unconscious bias on the part of the reporters (e.g., reaching out to men more often than to women, when a choice exists). We, of course, also know it is a result of societal bias. In a context where 71% of the Members of Parliament are male [74], it is natural to expect that we hear more often from male politicians. The fact that the current (in 2020) federal cabinet is gender-balanced probably helps. It does not, however, make up for the fact that the person at the top is a man. As shown in Table 4, Justin Trudeau, the Prime Minister, is quoted 8.3 times more often than Chrystia Freeland, arguably the most prominent woman politician in the country. At the top of the list of women is Bonnie Henry, the Public Health Officer for the province of British Columbia, a reflection of how important public health officers have become in the COVID-19 pandemic. And, clearly, Donald Trump is the

**Table 4. Top 15 quoted men and women in Canadian media between October 1, 2018 and September 30, 2020.**

| Identified as men | | | Identified as women | | |
|---|---|---|---|---|---|
| Name | # of quotes | Sector | Name | # of quotes | Sector |
| Donald Trump | 15,746 | Politics | Bonnie Henry | 2,239 | Public health |
| Justin Trudeau | 13,422 | Politics | Christine Elliott | 1,918 | Politics |
| Doug Ford | 6,760 | Politics | Chrystia Freeland | 1,890 | Politics |
| Jason Kenney | 4,190 | Politics | Nancy Pelosi | 1,718 | Politics |
| Andrew Scheer | 3,679 | Politics | Theresa Tam | 1,627 | Public health |
| François Legault | 2,754 | Politics | Jody Wilson Raybould | 1,493 | Politics |
| John Tory | 2,401 | Politics | Rachel Notley | 1,365 | Politics |
| Jagmeet Singh | 2,039 | Politics | Deena Hinshaw | 1,106 | Public health |
| John Horgan | 1,910 | Politics | Andrea Horwath | 1,053 | Politics |
| Joe Biden | 1,667 | Politics | Valérie Plante | 979 | Politics |
| Mike Pompeo | 1,661 | Politics | Patty Hajdu | 950 | Politics |
| Blaine Higgs | 1,659 | Politics | Catherine McKenna | 861 | Politics |
| Stephen McNeil | 1,595 | Politics | Meng Wanzhou | 681 | Private business |
| Boris Johnson | 1,553 | Politics | Elizabeth May | 671 | Politics |
| Scott Moe | 1,528 | Politics | Theresa May | 622 | Politics |
| **Total** | **62,564** | | **Total** | **19,173** | |

most quoted person by far in that time period. Perhaps the *style* of a person's statements, in addition to their content, makes the press more likely to find them quotable.

It is important at this point to emphasize that we do not distinguish between Justin Trudeau as a source of information (the way *source* is typically used by reporters) and Justin Trudeau saying something that reporters felt the need to quote, even if it is not new or privileged information, of the type that sources typically provide. To us, they are both instances of a 'quote', and Justin Trudeau is equally the source in both cases. Either of these cases is significant enough in that it points to reporters giving people who are already quoted frequently more of a voice.

The occupation of people quoted, as shown in Table 4, is also quite illuminating. Politics dominates, including international figures (Nancy Pelosi, Boris Johnson), and Canadian politicians at the federal (Elizabeth May, Jagmeet Singh), provincial (Rachel Notley, Doug Ford), and municipal (Valérie Plante, John Tory) levels. The diversity in the list of quoted women is perhaps more interesting. It includes Meng Wanzhou, Chief Financial Officer of Huawei, who was arrested in Vancouver in December 2018 and is in the middle of a legal extradition process as of 2020.

The three other female names that are not politicians are public health officers (Bonnie Henry, British Columbia's Public Health Officer; Theresa Tam, Chief Public Health Officer of Canada; and Deena Hinshaw, Alberta's Public Health Officer). One could also include Patty Hajdu (federal Minister of Health) and Christine Elliott (Ontario's Minister of Health) in the list of public health officers. All these women have been frequently quoted as a consequence of the COVID-19 pandemic, and started appearing in monthly top lists only in January 2020. By comparison, in the top 15 women quoted in December 2019 are Bonnie Lysyk, Auditor General of Ontario, who released her annual report that month, and environmental activist Greta Thunberg.

From these top-15 lists, it does seem that lack of equal representation in sources is partly due to lack of equal representation in society in general and in politics in particular. Indeed, political empowerment is the area where women are most underrepresented across the world

[4]. We do not believe, however, that news organizations and journalists are powerless to change the overall numbers we see on the Gender Gap Tracker dashboard. We know that the bias is pervasive and extends to expert sources and other areas where a choice does exist. Franks and Howell [75] discuss how the gender gap in broadcast media applies to prominent public figures and expert sources alike. They find that the source of the gap may be in who is hired and promoted within news organizations, with more men being hired, despite the fact that a larger number of women graduate from TV and broadcast university programs.

## Most frequent sources by category

In order to obtain a more extensive snapshot of who is being quoted by occupation, we conducted an annotation experiment. We extracted the top 100 men and women quoted for each of the 24 months between October 1, 2018 and September 30, 2020. We then manually annotated each of those sources and labelled them according to their occupation or the reason they were being quoted. The categories in Table 5 were based on previous work on manual source classification [27]. Most of the categories are self-explanatory. We assign 'Unelected government official' to cases such as attorneys general, government auditors, and (Canadian) governors, that is, cases where the person fills a political or representative role, but they were appointed, not elected. Health professionals can be considered unelected government officials (e.g., the Public Health Officer). However, given their prominence during COVID-19, we chose to assign them to a separate category, 'Health profession'. 'Perpetrators' may be accused (i.e., alleged perpetrators) or convicted. In 'Creative industries' we include artists, actors, and celebrities. Journalists and anchors are assigned to 'Media'. The category 'Person on the street interviews' is used for random interviews, or cases where the person is affected by an event (e.g., a flood), but cannot be considered a victim. 'Error' refers to cases where a name was wrongly identified as that of a person (e.g., Thomas Cook as a person). Errors in gender prediction are reported in Table 3.

**Table 5. Top 100 male/female sources, by category, in each of the 24 months between October 1, 2018 and September 30, 2020.**

| Category | Identified as men | | | | Identified as women | | | |
|---|---|---|---|---|---|---|---|---|
| | Quotes | | Unique persons | | Quotes | | Unique persons | |
| Politician | 103,378 | 73.8% | 295 | 40.4% | 29,007 | 52.6% | 270 | 24.7% |
| Sports | 10,723 | 7.7% | 113 | 15.5% | 1,415 | 2.6% | 60 | 5.5% |
| Unelected government official | 9,175 | 6.5% | 75 | 10.3% | 4,583 | 8.3% | 153 | 14.0% |
| Health profession | 5,327 | 3.8% | 21 | 2.9% | 9,217 | 16.7% | 58 | 5.3% |
| Leader (union, education, activist) | 2,297 | 1.6% | 36 | 4.9% | 1,578 | 2.9% | 65 | 6.0% |
| Police | 1,763 | 1.3% | 33 | 4.5% | 1,471 | 2.7% | 57 | 5.2% |
| Private business | 1,739 | 1.2% | 40 | 5.5% | 1,553 | 2.8% | 57 | 5.2% |
| Legal profession | 1,319 | 0.9% | 33 | 4.5% | 1,171 | 2.1% | 69 | 6.3% |
| Creative industries | 1,278 | 0.9% | 19 | 2.6% | 1,011 | 1.8% | 49 | 4.5% |
| Perpetrator | 948 | 0.7% | 18 | 2.5% | 264 | 0.5% | 17 | 1.6% |
| Academic/researcher | 685 | 0.5% | 15 | 2.1% | 835 | 1.5% | 45 | 4.1% |
| Victim/witness | 634 | 0.5% | 10 | 1.4% | 1,424 | 2.6% | 94 | 8.6% |
| Media | 530 | 0.4% | 11 | 1.5% | 391 | 0.7% | 21 | 1.9% |
| Non-governmental organization | 245 | 0.2% | 7 | 1.0% | 959 | 1.7% | 53 | 4.9% |
| Error | 91 | 0.1% | 4 | 0.5% | 8 | 0.0% | 1 | 0.1% |
| Person on the street interviews | 24 | 0.0% | 1 | 0.1% | 262 | 0.5% | 22 | 2.0% |
| **Total** | **140,156** | | **731** | | **55,149** | | **1,091** | |

There are some interesting observations with regard to Table 5. First of all, we notice that there are more women than men being quoted overall for this period (1,091 women vs. 731 men). That is, we see more variety in women in terms of the number of people being quoted. The difference in the number of quotes, however, is astounding: Men are quoted almost three times as often as women. That is, even though we hear from more women, we hear from men more often. This fact probably accounts, in large part, for the gap that we see on the Gender Gap Tracker dashboard, which counts unique quotes (not unique persons). Additionally, we see that there is a large difference between the most frequently quoted category and the second most frequently quoted. For men, number 1 is politicians (103,378) and number 2 is sports figures (10,723). For women, politicians are also at the top (29,007), with health professionals second to politicians (9,217). (Note that the rows are sorted by frequency of quotes for people identified as men).

These two findings, that much more space is given to men (in terms of number of quotes) and that much more space is given to the top category or occupation, point to a possible Pareto distribution [76], the principle that a large proportion of the resources is held by a small percentage of the population. Originally applied to wealth inequality, Pareto distributions have been found for the size of cities, internet traffic, scientific citations [77], and for the reward systems in science [78]. The related Pareto principle, also known as the 80-20 rule, preferential attachment, or the Matthew effect ('the rich get richer'), quantifies the difference in distribution (80% of the wealth held by 20% of the population). It seems that the main obstacle to hearing more from women in the media is a form of preferential treatment to those who already have a voice. This effect has been described as a winner-take-all distribution [24], in society in general and in news media in particular. We do have to bear in mind, however, that the numbers in Table 5 are based on the top 100 citations for men and women in each month. That is, they inherently capture the top of the Pareto distribution. Table 4 captures an even smaller fragment of that distribution, because it considers only the top 15 across the two-year period.

Finally, we would like to make some observations about the relative distribution of men and women by category. It is interesting to observe that, by number of unique persons, politicians seem to be close to parity (295 men and 270 women). There is a stark difference, again, in the number of quotes, that is, the number of times they were quoted: over 103,000 quotes by the 295 male politicians compared to just over 29,000 by the 270 female politicians. In other words, when we hear a quote by a politician, that politician is a man 78% of the time. The difference is even higher in sports. An interesting asymmetry is found between perpetrators (78% of the quotes by perpetrators are by men) and victims or witnesses (31% of the quotes by victims are men). Categories where women outnumber men both in terms of quotes and unique persons quoted in the category include health professionals, non-governmental organizations, and academics or researchers.

## The role of author gender

Now that we have established that the majority of quotes in news articles are from men, it would be interesting to check whether this bias has any correlation with the gender of the authors. Our hypothesis is that authors may prefer to feature and interview people of their gender, that is, articles written by female authors may contain a higher ratio of female sources compared to articles written by male authors.

In order to test this hypothesis, we first tagged the gender of the author or authors of each article, using the same name-gender services that we utilized for gender recognition on people and sources mentioned in texts. The process for cleaning up the author fields is described in the S1 Appendix, Section A.3.3. We then extracted statistics for female and male sources within

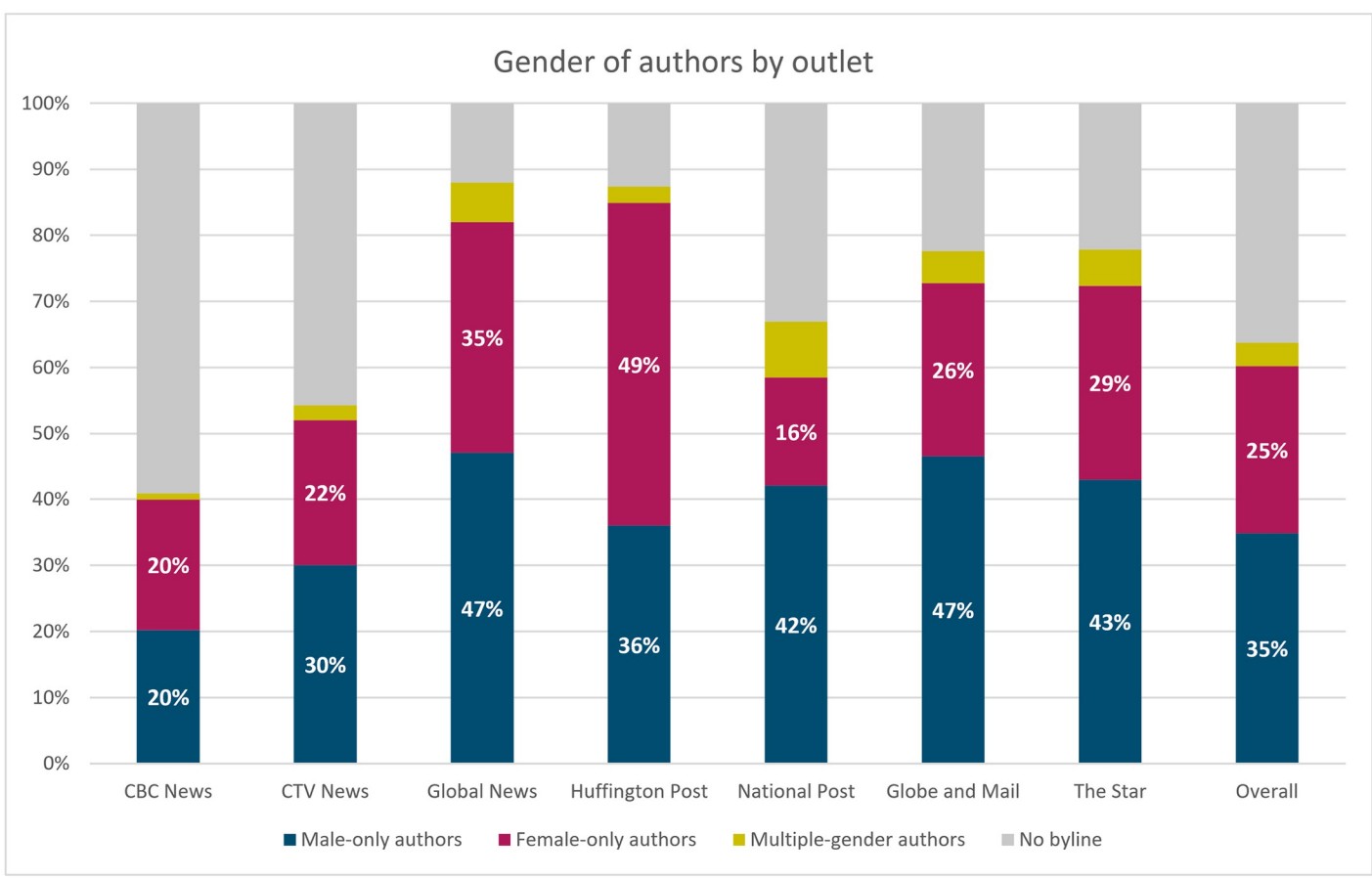

**Fig 3. Percentages of authors by gender, by outlet.** Dates: October 1, 2018 to September 30, 2020.

the publications of each news outlet, broken down into several categories: articles written by female authors only (155,197 articles), by male authors only (213,487 articles), by several authors of different genders (21,825 articles), and articles without a byline (222,041 articles). The last category encompasses different types of situations. It contains articles that had no byline or named author, such as editorials or newswire content. It also includes articles written by specific authors for which our system did not find a gender due to different limitations (e.g., the name does not exist in the gender databases). We know that our gender recognition services work quite well, because the rate of 'other' for sources mentioned (as opposed to authors) is quite low, at less than 1% for the entire period. Note also that this category is quite variable across news organizations. For instance, in the case of *CBC News*, where 'no byline' makes up the majority of the articles, this is because many articles do not have an author, but are posted as 'CBC News' or 'CBC Radio', or come from newswire sources.

From Fig 3, we see that, overall, the number of male authors exceeds the number of female authors in all outlets, except for *The Huffington Post* (49% women vs. 36% men, and 13% with no byline), which is also consistently the best performer in terms of female sources (see the line chart at the bottom of Fig 2).

Fig 4 shows, for each of the categories of authors described above, the percentage of times that they quoted female voices. The group at the bottom shows the aggregated percentage across all outlets, which speaks in favour of our hypothesis: Female authors are on average

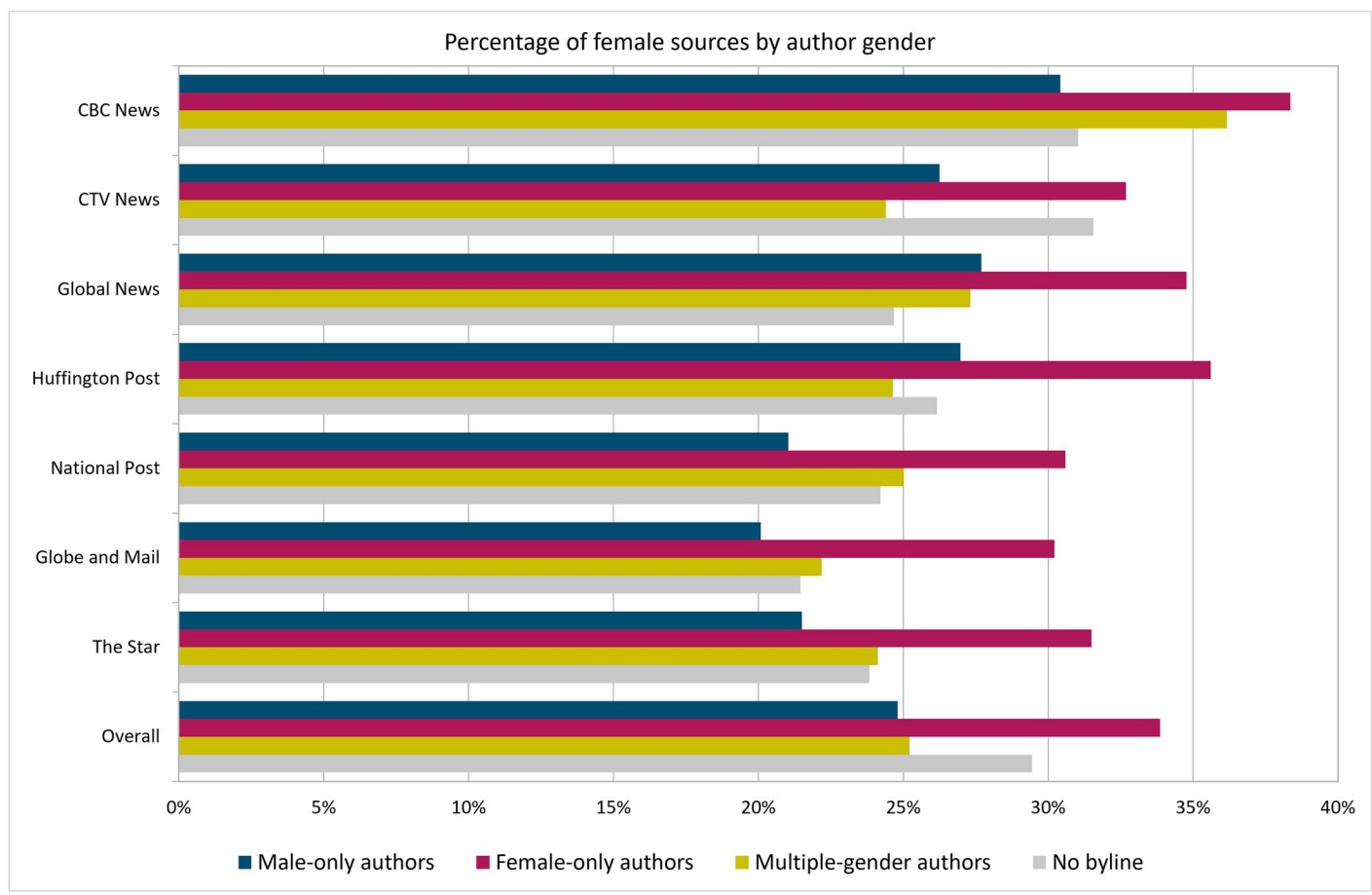

**Fig 4. Percentages of female sources across seven news outlets by author gender.** Dates: October 1, 2018 to September 30, 2020.

more likely than male authors to quote women in their articles. The chart shows that 34% of the sources are women in the articles authored by women, whereas this number is 25% in those authored by men.

Now let us examine the performance of male and female authors working for each of the news outlets. In all cases, without exception, articles written by women quote far more women than articles written by the other three groups. This suggests that part of the solution to the gender gap in media includes having more women reporters. This is true not only because women quote more women, but also because they seem to have a positive influence when part of a group. In most cases, articles written by a group that includes both men and women have more women quoted than articles written by men only. The two exceptions are *CTV News* (by a small margin, 26.3% women quoted with male-only authors vs. 24.4% with multiple genders) and *HuffPost* (by a slightly larger margin, 27.0% vs. 24.6%).

It is difficult to comment on the 'no byline' author category, as it includes many different types of authors, from editorials and newswire content to authors whose name we could not assign to a gender. In most cases, however, the trend is also that those articles tend to quote women more than articles under a male-only byline (with the exception of *Global News*, which also had the lowest percentage of articles without a byline).

In summary, the analyses in this section indicate that the bias towards quoting men seems to be strongest in articles written by men, a trend that has been observed in academic citations

**Table 6. Gender ratio in sources by article type.** Articles from *The Star* only. Dates: October 1, 2018 to March 31, 2020.

| Article type | Articles *n* | Male sources | Female sources | Other sources | Male sources % | Female sources % | Other sources % |
|---|---|---|---|---|---|---|---|
| In-house | 22,528 | 29,766 | 11,281 | 285 | 72.0% | 27.3% | 0.7% |
| Out-of-house | 13,400 | 17,359 | 6,246 | 182 | 73.0% | 26.3% | 0.8% |
| Newswire | 32,341 | 49,856 | 14,193 | 620 | 77.1% | 21.9% | 1.0% |

[79–83], Twitter mentions [84], including the Twitter circles of political journalists [85], and certainly in news articles [86]. Articles co-authored by a mix that includes male and female writers seem to contain a better balance of male and female sources of opinion; this observation points to collaboration between genders as a path towards closing the gender quote gap. This is, however, by no means a silver bullet. Recent analyses of the relationship between leadership in news organizations and balanced gender representation have found no correlation between the proportion of women producing the news and the proportion of women featured in the news [87]. This may have to do with a male-dominated culture in newsrooms, where professional identity overrides gender identity [87–89].

## The role of out-of-house content

One objection that news organizations may have is that, in some cases, they have no control over the breakdown of sources in an article, because they republish content, either from newswire or from other news publishers. Thus, it would be interesting to know whether there is a difference in the ratios depending on the source of the article.

As it turns out, classifying articles by source is a rather difficult task. We rely on the data we obtain by scraping, and specifically the author field. Unfortunately, the author field in an article does not always clearly indicate the author's affiliation or the source. We restricted our analyses to *The Toronto Star*, because that organization had expressed an interest in a more fine-grained analysis. Note that this analysis is for slightly different dates, the 18 months between October 2018 and March 2020.

Using a combination of patterns and regular expression searches, we classified all the articles of *The Star* into three categories: in-house, out-of-house, or newswire. Out-of-house articles were labelled using an extensive list of external publishers that *The Star* re-publishes (e.g., *LA Times, Washington Post*, and *Wall Street Journal*). Newswire articles were determined to originate from a handful of news agencies: Canadian Press, Associated Press, Bloomberg, and Reuters. We were careful to restrict our pattern matching to author fields, as articles written in-house sometimes contain photos from newswire organizations.

Using this method, we obtained the results in Table 6. (Note that our method has a margin of error: In a manually labelled sample of 10,000 articles, we found an error rate of 4.41%, almost always in the in-house articles. That is, articles that are out-of-house or newswire may be incorrectly identified as in-house.) We find that, regardless of the origin of the article, men are the dominant source, and that the proportions are quite similar for out-of-house and newswire content. It is encouraging, however, to see that articles written by *The Star* reporters are more inclusive than those originating outside the organization. *The Star* has publicly stated that they want to improve the proportion of female sources that they quote [90], and it seems to be the case that their reporters do better, even if the proportion is still far from parity.

## Conclusion

The main goal of the Gender Gap Tracker database and dashboard is to motivate news outlets to diversify their sources. This applies to all forms of diversity. While the Gender Gap Tracker

can only capture one kind of diversity, because it relies on names to assign gender to sources, we believe that other forms of diversity should be considered, as we know that many other groups are underrepresented in the news [91–96].

Gender equality is one of the United Nations' 17 Sustainable Development Goals [30]. We are, sadly, far from achieving gender equality in many areas of our societies. Gender representation in the media is, however, within our reach, if enough effort is devoted to this goal and if we incorporate accountability into the effort. We hope that the Gender Gap Tracker provides the type of accountability tool that will encourage and facilitate gender parity in sources. Two results from our analyses that we would like to highlight here suggest a path towards equality.

First of all, we saw in Fig 4 that articles by authors of multiple genders tend to quote women more often. That is, when the author list is diverse, so are the sources quoted. Other research suggests that diversity at the top, in editors and publishers, also has a positive effect on the proportion of women mentioned in the news [24], although it is not sufficient to have parity in the newsroom or increased female leadership in news organizations [87, 88, 97]. The relationship between female leadership and improved representation for women in the news is indeed quite complex [98].

Second, results (Tables 4 and 5) point to a lack of equality in how many times men and women are quoted overall, not just in how many men and women are quoted. Thus, although we see a certain tokenism in having female voices present in the news, their voices are drowned out by the overwhelming number of times that we hear from men, often from just a handful of men. It looks like women are given a presence, but then men get the majority of the space. This also points to a concentration of power at the top, which can be balanced by diversifying sources in general.

Journalists report that it takes more time and effort to reach diverse sources. There are many barriers for women to participate in civil society, and in particular for engaging with the media. One particularly harrowing issue that needs to be addressed is the abuse and harassment that women experience when they speak publicly, especially when they speak to controversial topics [99–103]. Women who engage in online discussions experience trolling, abusive comments, death and rape threats, and also threatening offline encounters, such as name-calling and public abuse [104, 105]. Jane [106] argues that the extent of the harassment online has offline consequences for women, which are manifested socially, psychologically, financially, and politically. Many women, understandably, self-censor to avoid such consequences. True equal representation in public discourse will be much more difficult to achieve if the rewards and consequences of participating are unequal across genders.

The size and richness of the data in the Gender Gap Tracker database lends itself to many interesting further analyses. One area that we are investigating is the relationship between the topic of the article and the gender of those quoted. The research question, simply put, is whether men are quoted more in financial news and women in arts and lifestyle articles. Our preliminary answer is that, indeed, this is the case, with a bright spot in the prominence of female voices in healthcare during the COVID-19 pandemic [107]. Topic-based analyses can also help identify emerging topics, such as one-time events (terrorist attacks, sports events) or new developments that stay in the news (Brexit, COVID-19).

We have also informally explored the relative prominence of political candidates in several elections [108]. We found that eventual winners of elections were more likely to be quoted in the period leading up to the election in most elections we studied (but not all). We, of course, do not propose a causal relation between presence in the media and likelihood of being elected. Even if there is a causal relation, the cause and effect direction is unclear. It could be that the more well-known the candidate, the higher their chances of being elected. It could also be the case that when a candidate seems to be leading in the polls, they are more likely to be quoted

in the news media. Further analyses as new elections take place would shed more light onto those questions.

Other research avenues that could be pursued relate to questions of salience and space, i.e., whether quotes by men are presented more prominently in the article, and whether men are given more space (perhaps counted in number of words). Finally, more nuanced questions that involve language analysis include whether the quotes are presented differently in terms of endorsement or distance from the content of the quote (*stated* vs. *claimed*). We plan to pursue some of those questions, but also invite researchers to join in this effort. The data collected for this project can be made available, upon request, for non-commercial research purposes.

## Data and code

The data was downloaded from public and subscription websites of newspapers, under the 'fair dealing' provision in Canada's Copyright Act. This means that the data can be made available only for private study and/or research purposes, and not for commercial purposes. As such, the data will be made available upon request and after signing a license agreement. Contact for data access: Maite Taboada (mtaboada@sfu.ca) or Research Computing Group at Simon Fraser University (research-support@sfu.ca).

The code is available on GitHub under a GNU General Public License (v3.0). The authors of this paper are the creators of the code and own the copyright to it: https://github.com/sfu-discourse-lab/GenderGapTracker.

A light-weight version of the NLP module is also made available for processing one article at a time: https://gendergaptracker.research.sfu.ca/apps/textanalyzer.

## Supporting information

**S1 Appendix.**
(PDF)

## Acknowledgments

The Gender Gap Tracker is a collaboration between the Discourse Processing Lab, the Big Data Initiative at Simon Fraser University, and Informed Opinions. Our thanks and admiration to Shari Graydon of Informed Opinions for initiating this project and for being a tireless advocate for gender equality. We would like to thank Kelly Nolan, Dugan O'Neil, and John Simpson for bringing us together, and John especially for the initial design of the database. Yanlin An, Danyi Huang, and Nilan Saha contributed to the heuristic quote extraction process, as part of a capstone project in the Master of Data Science program at the University of British Columbia. Thank you to members of the Discourse Processing Lab at SFU for feedback, insight, and help with evaluation: Laurens Bosman, Lucas Chambers, Katharina Ehret, Rohan Ben Joseph, and Varada Kolhatkar. Special thanks to Lucas Chambers for tracking down references and for editing assistance.

## Author Contributions

**Conceptualization:** Fatemeh Torabi Asr.

**Data curation:** Mohammad Mazraeh, Alexandre Lopes, Vagrant Gautam, Prashanth Rao.

**Formal analysis:** Fatemeh Torabi Asr, Mohammad Mazraeh, Prashanth Rao, Maite Taboada.

**Funding acquisition:** Maite Taboada.

**Investigation:** Fatemeh Torabi Asr, Mohammad Mazraeh, Alexandre Lopes, Vagrant Gautam, Maite Taboada.

**Methodology:** Fatemeh Torabi Asr, Mohammad Mazraeh, Alexandre Lopes, Vagrant Gautam, Maite Taboada.

**Project administration:** Fatemeh Torabi Asr, Maite Taboada.

**Resources:** Alexandre Lopes, Maite Taboada.

**Software:** Fatemeh Torabi Asr, Mohammad Mazraeh, Alexandre Lopes, Vagrant Gautam, Prashanth Rao, Maite Taboada.

**Supervision:** Fatemeh Torabi Asr, Maite Taboada.

**Validation:** Fatemeh Torabi Asr, Mohammad Mazraeh, Alexandre Lopes, Vagrant Gautam, Junette Gonzales, Prashanth Rao, Maite Taboada.

**Visualization:** Fatemeh Torabi Asr, Prashanth Rao, Maite Taboada.

**Writing – original draft:** Fatemeh Torabi Asr, Mohammad Mazraeh, Vagrant Gautam, Junette Gonzales, Maite Taboada.

**Writing – review & editing:** Fatemeh Torabi Asr, Mohammad Mazraeh, Vagrant Gautam, Junette Gonzales, Prashanth Rao, Maite Taboada.

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
