## [Decision Letter · Decision Letter 0]

9 Oct 2020

PONE-D-20-22487

The Gender Gap Tracker: Using Natural Language Processing to measure gender bias in media

PLOS ONE

Dear Dr. Taboada,

Thank you for submitting your manuscript to PLOS ONE. After careful consideration, we feel that it has merit but does not fully meet PLOS ONE’s publication criteria as it currently stands. Therefore, we invite you to submit a revised version of the manuscript that addresses the points raised during the review process.

Two expert reviewers have weighed in on the manuscript, and I have read it carefully myself.  I found the manuscript to be very clearly written and easy to read, and I believe we all would concur that this represents a substantial piece of research that addresses an important societal problem.  That having been said, the reviewers and I also have a number of comments, criticisms, and questions, some of which are summarized below.  I am following the recommendation of both reviewers in adopting a Major Revision decision. 

Reviewer 1 remarks that the paper is unnecessarily lengthy and times loses its focus.  I agree.  In general, I would discourage you from overly elaborating on purely implementational details (e.g., what python libraries were used; Section 3), approaches you considered but didn’t pursue (e.g., the section on pronoun-based gender prediction), narrated walkthroughs of either your research process (e.g., Section 4.3) or particular algorithms (e.g., Sections 3, 4.2), and data structures (e.g., Figure 4) unless there is a compelling reason why readers need to be apprised of such details (in which case one could still strive for brevity).  The reviewer cites several other places that in their view were overly detailed.  I’ll also add that whereas I think the discussion of issues regarding binary notions of gender is important, I also think it suffices to have this discussion occur just once, presumably early in the paper (discussions currently occur in Sections 1, 2.3, and 4.2.)  

Reviewer 2 points out that the paper would benefit from a more rigorous analysis to evaluate the question under scrutiny while controlling for other factors.  I concur here as well.  This reviewer provides the popularity of an individual as an example.  Adding to this, it seems apparent that there are a number of reasons a quote might appear in a news outlet, some of which seem orthogonal to the question of gender bias in news sources.  For instance, the submission notes that Donald Trump is the most quoted person by far, but it seems doubtful that he’s being used as a source in many of these cases; it is news itself when the US President speaks.  If Hillary Clinton was the current US President instead, the numbers would presumably be quite different, but that fact seems incidental to the question.  Further, as the submission acknowledges, journalists often don’t have a choice when citing a source -- for example, if one seeks a quote from the Police Chief of a certain area (say, where riots are occurring), one has no control over that person’s gender.  Section 7 includes interesting discussions of such issues that in my view add considerable value to the paper, but given that “the main goal of the database and the dashboard is to motivate news outlets to diversify their sources” (first line of the conclusion), it remains unclear to me how the various confounds can be sorted through so as to yield a more fair and actionable measurement of media bias.  I’m not sure of the precise remedy here, but would nonetheless encourage any additional analyses the authors might offer. 

Finally, a small comment, regarding the evaluation of quote extraction on page 20 – to be a valid evaluation, I would think that the corpus would have to be distinct from the larger corpus used in the development process outlined at the top of Section 6, but it’s not obvious to me that it was.  So this could use clarification.  (An even smaller point – I didn’t understand the “at least” part in the first line of page 19, if the corpus totaled 98 documents). 

Although we cannot accept your submission in its current form, the reviewers and I are agreed that there is considerable value in your submission, and hence I would encourage you to submit a revised version of your manuscript if you're inclined to do so after taking on a thorough consideration of the aforementioned feedback and other comments made by the reviewers.  Upon resubmission, I would ask the same two reviewers to evaluate a resubmission, only recruiting new ones if one or both were to decline.

We look forward to receiving your revised manuscript.

Kind regards,

Andrew Kehler, Ph.D

Academic Editor

PLOS ONE

Journal Requirements:

3. We note that Figures [2, 6] in your submission contain copyrighted images. All PLOS content is published under the Creative Commons Attribution License (CC BY 4.0), which means that the manuscript, images, and Supporting Information files will be freely available online, and any third party is permitted to access, download, copy, distribute, and use these materials in any way, even commercially, with proper attribution. For more information, see our copyright guidelines: http://journals.plos.org/plosone/s/licenses-and-copyright.

1.         You may seek permission from the original copyright holder of Figure(s) [2 and 6] to publish the content specifically under the CC BY 4.0 license.

4. We note that Figure [2] includes an image of a participant  in the study. 

Reviewers' comments:

Reviewer's Responses to Questions

**Comments to the Author**

1. Is the manuscript technically sound, and do the data support the conclusions?

Reviewer #1: Yes

Reviewer #2: Partly

2. Has the statistical analysis been performed appropriately and rigorously? 

Reviewer #1: Yes

Reviewer #2: Yes

3. Have the authors made all data underlying the findings in their manuscript fully available?

Reviewer #1: Yes

Reviewer #2: Yes

4. Is the manuscript presented in an intelligible fashion and written in standard English?

Reviewer #1: Yes

Reviewer #2: Yes

5. Review Comments to the Author

Reviewer #1: This paper investigates gender bias in media by analyzing the number of men and women quoted in Canadian news texts. Additionally, the authors have developed and made publicly available a tool Gender Gap Tracker that enables tracking of daily publications from a number of Canadian news websites. Prior work was manual to high extent and has not addressed both direct and indirect quotes on a continuous basis.

Overall comments:

- Well written and researched introduction

- Because of some lengthy sections the paper looses its focus. A lot of NLP and CS concepts are explained in very much detail. Evaluation takes as much space as the actual results and their analysis. Suggestion: Make use of Appendix

- Some results are presented already in the methodological chapters, whereas some concepts that should be mentioned in methodology are described in the middle of evaluation.

- I would wish for a stronger conclusion chapter to point out the most important take aways.

More detailed comments:

- The paper in general is unnecessarily lengthy. Some concepts described in the section related work are repeated again in the methods section. For instance, in chapter 4 a quite long description of NeuralCoref, in chapter 4.2 a very long description of the gender prediction process, in section 4.3 the explanation why regular expressions are inadequate, or description of Figure 8.

- (Data Scraping, Lines 299-300): Have you tried parallelization? Even small parallelisation might considerably speed up the process without overloading the website.

- (Identifying people and predicting their gender/Quotation extraction): Swapping these two chapters feels more natural. Quotation extraction is the first step before people can be identified.

- (Identifying people and predicting their gender): In section 4.2 about Name-based gender prediction it is mentioned that web services are used and the errors are corrected when encountered. When is it tested? Link to the results from chapter 6?

- (Identifying people and predicting their gender, Lines 439-441): Link to the results from chapter 6?

- (Identifying people and predicting their gender): Section 4.3 list of author names containing major organisation names, any attempt to create it automatically? Add the manually created list to the Appendix?

- (Quotation extraction, 5.1): What are the attempts to include “according to” quotes? What was the reason/challenge that they are not included now?

- (Quotation extraction, 5.2): Manually investigated 10 articles - with how many quotes? More details? Shouldn’t it be covered in the evaluation section and not methods?

- (Evaluation, 6.2) Annotations only on 8 articles and 3 annotators with 1 final? Is it enough?

- (Evaluation, 6.3) Description of JSON format and annotator's challanges to be shortened.

- (Evaluation, 6.4) Concepts like scare quotes and what counts as a quote should be explained earlier in Section 5

- (Evaluation, 6.5, Lines 810-812) How were the threshold values chosen? Any previous testing in this regard or arbitrary?

- (Analysis and observation, 7.2, Lines 945-947): The professions can be also extracted from wiki to save manual labor

Reviewer #2: The paper attempts to characterize gender bias in media reporting by a large scale quantitative analysis of quoting patterns of seven Canadian outlets. To achieve this, the authors present a natural language processing system that applies various natural language processing (NLP) techniques to drive the above analysis which suggests that there is still a significant gender gap biased against women in media reporting. The research questions investigated are of immense importance to society and public policy and will play an important role in efforts undertaken to improve diversity in organizations and society. Yet another strength of the paper is an attempt to characterize the accuracy of the various NLP methods before applying them to answer the research question. Broadly, the methods used appears reasonable and the conclusions are inline with observations made in prior work. While the NLP system built to computationally extract named entities and quoting patterns looks pretty solid, several concerns arise in the subsequent analyses: In particular, the paper could significantly benefit by adopting a more rigorous analysis approach to ascertain the bias while controlling for various factors (using linear fixed effects models). For example, what is the effect of popularity of the individual (number of mentions of name) on the gender gap? In particular, a very related work by Shor et.al (2015) also conducted a very similar analysis where they observe that when covering famous individuals the gender gap in printed media coverage observed is much larger than the coverage of no t so popular individuals (see Figure 2 of Shor et. al (2015)). More broadly, the paper also does not justify its focus on only looking at reported speech (when male or female entities are quotes) as opposed to just looking at all mentions of people (this was done by Shor et. al (2015)). It could be possible that people could be covered but they are not quoted (either syntactically or directly). For example, Serena Williams won the championship mentions (gives media coverage to Serena) "Serena Williams" but does not quote her. In general, the paper would benefit by placing their results and analysis in context and how they relate to findings presented in Shor et. al(2015). Similarly when presenting the analysis of role of gender (by outlet), it would be useful to control for other aspects like (newspaper section etc).

Line 288: Why is 1-b not considered a valid article/url? It Seems that the cut-off of 5 directories is arbitrary.

Also does the index page of the media outlet contain all the articles that were published that day or just the top X? If just the top X then there might be a coverage issue of the data (since articles crawled would be biased towards those stories/articles that are perceived by editor to boost viewership count).

References:

Shor, Eran, et al. "A Paper ceiling: Explaining the persistent underrepresentation of women in printed news." American Sociological Review 80.5 (2015): 960-984.

6. PLOS authors have the option to publish the peer review history of their article (what does this mean?). If published, this will include your full peer review and any attached files.

Reviewer #1: No

Reviewer #2: No

---

## [Decision Letter · Decision Letter 1]

3 Dec 2020

PONE-D-20-22487R1

The Gender Gap Tracker: Using Natural Language Processing to measure gender bias in media

PLOS ONE

Dear Dr. Taboada,

Thank you for submitting your revised manuscript to PLOS ONE.  Upon receipt of the submission, I requested reviews from the original two reviewers; Reviewer 2 accepted but Reviewer 1 declined.  Based on my own reading of the revision, I felt comfortable basing my decision on the judgment of Reviewer 2 and my own, and hence opted not to bring a new reviewer into the process. 

Reviewer 2 and I agree that the original comments from both reviewers have been acted on in good faith, and that the paper is publishable.  Because submissions that receive Accept decisions at the journal proceed straight to production, I'm taking the action of issuing a Minor Revision decision, to give you the opportunity to address the minor comments that Reviewer 2 makes in their new review, as well as any other minor modifications that you deem appropriate before the article sees print.  Barring any new substantial changes, I intend to accept the revised manuscript upon a spot check.  Hence it will not go out for further external review. 

We look forward to receiving your revised manuscript.

Kind regards,

Andrew Kehler, Ph.D

Academic Editor

PLOS ONE

Reviewers' comments:

Reviewer's Responses to Questions

**Comments to the Author**

1. If the authors have adequately addressed your comments raised in a previous round of review and you feel that this manuscript is now acceptable for publication, you may indicate that here to bypass the “Comments to the Author” section, enter your conflict of interest statement in the “Confidential to Editor” section, and submit your "Accept" recommendation.

Reviewer #2: (No Response)

2. Is the manuscript technically sound, and do the data support the conclusions?

Reviewer #2: Yes

3. Has the statistical analysis been performed appropriately and rigorously? 

Reviewer #2: Yes

4. Have the authors made all data underlying the findings in their manuscript fully available?

Reviewer #2: Yes

5. Is the manuscript presented in an intelligible fashion and written in standard English?

Reviewer #2: Yes

6. Review Comments to the Author

Reviewer #2: The revised version of the paper addresses most of the comments I had in the previous version satisfactorily. Since the authors defer the more rigorous/deeper analysis to future work (including topical analysis etc.) and the primary contribution here is the Gender Gap Tracker software, I would encourage the authors to emphasize that here they are primarily interested in demonstrating the rich analyses that the Gender Tracker would enable in the future (and some of the analyses may be strengthened further as noted in my previous comments). Finally, one point that the authors mention the response letter is that to prevent overfitting due to iterations over their rules, they ensured they always evaluated on an extension of the test set. This detail is not mentioned in the revised paper's main content -- an important point which should be added.

7. PLOS authors have the option to publish the peer review history of their article (what does this mean?). If published, this will include your full peer review and any attached files.

Reviewer #2: No

---

## [Author Response · Author response to Decision Letter 1]

23 Dec 2020

Please see response to reviewers letter.

---

## [Editor Report · Decision Letter 2]

4 Jan 2021

The Gender Gap Tracker: Using Natural Language Processing to measure gender bias in media

PONE-D-20-22487R2

Dear Dr. Taboada,

We’re pleased to inform you that your manuscript has been judged scientifically suitable for publication and will be formally accepted for publication once it meets all outstanding technical requirements.

Kind regards,

Andrew Kehler, Ph.D

Academic Editor

PLOS ONE

---

## [Editor Report · Acceptance letter]

8 Jan 2021

PONE-D-20-22487R2 

The Gender Gap Tracker: Using Natural Language Processing to measure gender bias
in media 

Dear Dr. Taboada:

I'm pleased to inform you that your manuscript has been deemed suitable for publication in PLOS ONE. Congratulations! Your manuscript is now with our production department. 

Kind regards, 

on behalf of

Dr. Andrew Kehler 

Academic Editor

PLOS ONE